# Unlocking Zero-shot Potential of Semi-dense Image Matching via Gaussian Splatting

## Abstract

Learning-based image matching critically depends on large-scale, diverse, and geometrically accurate training data. 3D Gaussian Splatting (3DGS) enables photo-realistic novel-view synthesis and thus is attractive for data generation. However, its geometric inaccuracies and biased depth rendering currently prevent robust correspondence labeling. To address this, we introduce **MatchGS**, the first framework designed to systematically correct and leverage 3DGS for robust, zero-shot image matching. Our approach is twofold: (1) a **geometrically-faithful data generation pipeline** that refines 3DGS geometry to produce highly precise correspondence labels, enabling the synthesis of a vast and diverse range of viewpoints without compromising rendering fidelity; and (2) a **2D-3D representation alignment strategy** that infuses 3DGS' explicit 3D knowledge into the 2D matcher, guiding 2D semi-dense matchers to learn viewpoint-invariant 3D representations. Our generated ground-truth correspondences reduce the epipolar error by up to 40 times compared to existing datasets, enable supervision under extreme viewpoint changes, and provide self-supervisory signals through Gaussian attributes. Consequently, state-of-the-art matchers trained solely on our data achieve significant zero-shot performance gains on public benchmarks, with improvements of up to 17.7%. Our work demonstrates that with proper geometric refinement, 3DGS can serve as a scalable, high-fidelity, and structurally-rich data source, paving the way for a new generation of robust zero-shot image matchers.

## 1 Introduction

Reliable pixel-level correspondences are fundamental to modern 3D vision, supporting applications from classical Structure-from-Motion (SfM) (Schonberger & Frahm, 2016) and SLAM (Campos et al., 2021) to recent advances in 4D reconstruction (Jin et al., 2024; Chen et al., 2025) and radiance field rendering (Mildenhall et al., 2021; Kerbl et al., 2023). This task of image matching has seen a paradigm shift from hand-crafted methods like SIFT (Lowe, 2004) to learning-based approaches such as SuperGlue (Sarlin et al., 2020) and LoFTR (Sun et al., 2021), which now define the state-of-the-art. However, the success of deep learning approaches critically depends on the scale, diversity, and accuracy of their training data.

For years, datasets like ScanNet (Dai et al., 2017a) and MegaDepth (Li & Snavely, 2018), captured with depth sensors or reconstructed via SfM, have been the primary sources for geometric supervision. Despite their quality, they are limited in scene and viewpoint diversity. Recent efforts like GIM (Shen et al., 2024) and L2M (Liang et al., 2025) have sought to expand data availability by generating pseudo or synthetic labels from large-scale video or image collections. While increasing data volume, their sampled viewpoints remain constrained by photographers' views and lack the global geometric consistency of a fully reconstructed 3D scene. Such dense and globally consistent supervision remains invaluable, as it provides the unambiguous, geometrically-grounded signal necessary for learning a coherent matching policy robust to variety on viewpoint and surface texture.

Recently, **3D Gaussian Splatting (3DGS)** (Kerbl et al., 2023) has demonstrated strong capabilities in high-fidelity novel view synthesis. It is naturally suited as a data generation pipeline for image matching due to its support for **free-viewpoint sampling**. From a reconstructed 3DGS scene, we can generate a virtually infinite dataset by freely controlling camera poses, intrinsics, and inter-frame overlap. This allows for the targeted synthesis of challenging cases, such as extreme viewpoint, large zoom-in/zoom-out variations, and very low overlap, that are rare in real-world datasets and crucial for improving model robustness. Furthermore, the Gaussian primitives provide an **explicit 3D representation**, opening the door to training image matchers that are inherently 3D-aware.

Figure 1: **(a) illustrates our data generation pipeline.** Given train-view images and monocular depth priors, we first reconstruct the scene using our geometry-improved 3DGS. Augmented viewpoints are then generated from train views, with pre-rendering checks removing outliers before rendering usable data. **(b-1) to (b-4) compares four depth rendering methods detailed in Sec. 3.1.**

However, leveraging 3DGS for geometrically precise annotations is non-trivial. As recent work Guédon & Lepetit (2024); Jiang et al. (2024) points out, the Gaussian primitives are optimized for rendering quality, not geometric fidelity. Consequently, they often fail to adhere to the true scene surface. This geometric inaccuracy is compounded by biased depth maps from the default alpha-blending renderer. Together, these issues result in significant errors in the final correspondence labels, including mismatches and missing pairs.

To address these challenges, we introduce **MatchGS**, a framework designed to unlock the full potential of 3DGS for zero-shot image matching. Our solution is twofold. First, we propose a **geometrically-faithful data generation pipeline** that significantly enhances the precision of standard 3DGS (Fig. 1). Through systematic geometric corrections and regularization, we produce dense, accurate, and unbiased correspondence labels suitable for robust training. Second, we introduce a **2D-3D representation alignment strategy** that infuses 3DGS' explicit 3D knowledge into the 2D matcher (Fig. 3). This derives from attempts at two complementary scales: a contrastive objective aligns 2D patch features with 3D voxel representations at the coarse scale, while direct attribute regression guides fine-level matching at the pixel scale.

Our pipeline efficiently generates vast and reliable training data (visualized in Fig. 2), combining the geometric consistency of a full 3D scene with expansive viewpoint diversity. Furthermore, it is readily scalable to large-scale multi-view datasets (Ling et al., 2024), enabling broad scene diversity. Simultaneously, our 2D-3D alignment endows the matcher with viewpoint-invariant 3D representations, significantly enhancing its robustness to unseen scenes and viewpoint changes. We find this is most effective when aligning at a coarse, patch-to-voxel scale, which provides a more stable 3D representation than a noisy pixel-to-primitive mapping.

Extensive experiments validate the effectiveness of MatchGS. First, our generated annotations exhibit superior geometric precision, reducing epipolar error by 40 times compared to standard datasets (Li & Snavely, 2018; Dai et al., 2017a). Second, existing matchers trained with MatchGS achieve significant zero-shot performance gains on public benchmarks. Compared to their baselines trained on MegaDepth, ELoFTR (Wang et al., 2024) improves by 16.2% on ZEB (Shen et al., 2024) and 13.9% on ScanNet, while LoFTR (Sun et al., 2021) improves by 11.2% on ZEB and 17.7% on ScanNet. Our contributions are summarized as follows:

- A High-Fidelity Data Generation Pipeline. Our pipeline corrects 3DGS' geometry to produce reliable and dense correspondences, particularly for challenging conditions like large viewpoint changes that are hard to collect in existing image matching datasets.

- A 2D-3D Representation Alignment Strategy. We leverage explicit 3D attributes from the 3DGS scene to guide 2D image matchers, resulting in representations that are significantly more robust to viewpoint changes and yield better zero-shot performance.

- Effective Zero-Shot Generalization. Our experiments show that image matching models trained solely on our data achieve substantial improvements in generalization, outperforming state-of-the-art baselines on multiple public benchmarks.

Figure 2: **Visualization of data generation quality.** Our proposed pipeline can freely generate dense and accurate labels under large variations in viewpoint and scale.

## 2 RELATED WORK

**Image matching datasets.** MegaDepth (Li & Snavely, 2018) reconstructs 196 Internet scenes with COLMAP (Schonberger & Frahm, 2016), but its depth maps remain incomplete and noisy despite MVS and semantic refinements, causing boundary errors and unreliable ground-truth sampling. ScanNet (Dai et al., 2017a) reconstructs 1613 indoor scenes with RGBD sensors and BundleFusion (Dai et al., 2017b), ensuring global geometric consistency but requiring physical scene scanning with dedicated devices. Beyond reconstruction, GIM (Shen et al., 2024) generates pseudo labels from Internet videos with pretrained matchers and temporal propagation, turning hundreds of hours of videos into potential supervision. But accumulated errors lead MAGSAC (Barath et al., 2019) to discard many pairs, which results in gradually sparse label density. While dynamic occlusions further undermine propagation reliability. L2M (Liang et al., 2025) lifts 2D images to colored point clouds and inpaints novel views to form multi-view pairs. While abundant image collections provide scene diversity, simple point cloud reprojection cannot ensure synthesis fidelity, and inpainting fails under large pose shifts or complex occlusions, limiting the simulation of wide baselines and extreme views. Overall, existing approaches have yet to simultaneously achieve global geometric consistency, which enables dense and reliable supervision across large baselines, and sampling diversity, which supports generalization to new viewpoints and scenes. Our pipeline addresses both aspects by providing scalable scene expansion with consistent geometry and diverse viewpoints.

**Image matching methods.** Traditional pipelines involve keypoint detection, descriptor extraction, and matching. Hand-crafted methods such as SIFT (Lowe, 2004) and ORB (Rublee et al., 2011) follow this paradigm and remain widely used in SfM and SLAM. SuperPoint (DeTone et al., 2018), extending MagicPoint (DeTone et al., 2017), introduced self-supervised joint detection and description via homography adaptation. SuperGlue (Sarlin et al., 2020) further modeled context-aware correspondences with a graph neural network, setting a strong benchmark for sparse matching. LoFTR (Sun et al., 2021) pioneered detector-free dense correspondence learning with Transformers (Vaswani et al., 2017), enabling reliable matches even in low-texture regions. DKM (Edstedt et al., 2023) later showed that dense methods can excel in two-view geometry, achieving state-of-the-art results. While most methods optimize for in-domain datasets, hand-crafted RootSIFT (Arandjelović & Zisserman, 2012) continues to perform competitively in the wild (Jin et al., 2021; Shen et al., 2024), motivating greater focus on zero-shot generalization. GIM and L2M enhance generalization by scaling scene diversity, whereas we show that even with limited scenes, free viewpoint sampling and viewpoint-invariant 3D representations can substantially improve the zero-shot performance of semi-dense matching models.

**Representation alignment.** Representation alignment has been explored across multiple domains. CLIP (Radford et al., 2021) uses contrastive learning to align images and text in a shared space, enabling strong zero-shot transfer. REPA (Yu et al., 2024) aligns hidden states of a diffusion model with clean image features from a pretrained encoder, improving both training efficiency and generative quality. In 3D vision, 3DG-STFM (Mao et al., 2022) transfers RGB-D knowledge to RGB via distillation to enhance feature matching. FiT3D (Yue et al., 2024) fine-tunes 2D backbones with features rendered from 3D Gaussian splatting, while L2M (Liang et al., 2025) supervises encoders with rendered Gaussian maps for multi-view perception. These methods leverage 3D information to supervise model weight updates, thereby implicitly encouraging the model to learn 3D-aware features. By comparison, our approach constructs a consistent embedding in a unified 2D-3D representation space, which directly affecting the correlation matrix and mutual nearest-neighbor matching.

# 3 METHODOLOGY

In this section, we systematically investigate how to extend 3D Gaussian Splatting (3DGS) (Kerbl et al., 2023) into a training framework for image matching. This framework includes a data generation pipeline for dense and accurate supervision signals, and a representation alignment strategy for additional self-supervisory signals. Our discussion is centered around two core questions:

***Q1:*** *Is it feasible to design a data pipeline that relies solely on 3DGS for zero-shot image matching, without requiring additional pre-training or fine-tuning?*

**Answer:** We show that by improving the depth rendering quality and controlling the sampling of novel viewpoints, 3DGS can be leveraged to generate high-fidelity image pairs and dense annotations for challenging samples. This lays the foundation of our zero-shot training framework, which we detail in Sec. 3.1.

***Q2:*** *Given that the framework already provides high-quality image pairs and annotations, can we further exploit the explicit attributes of gaussian primitives to guide 2D semi-dense matching models to learn viewpoint-invariant 3D representations?*

**Answer:** We investigate how to incorporate 2D-3D representation alignment to exploit Gaussian attributes for viewpoint-invariant aware semi-dense image matching. Two paradigms are explored to enhance model representations from different perspectives, as described in Sec. 3.2.

## 3.1 UNLOCKING FREE-VIEWPOINT DATA GENERATION

To obtain reliable image matching annotations from 3DGS, two conditions are essential: (1) accurate geometry for depth-based correspondence generation, and (2) photorealistic novel views to minimize distribution gaps with real images. We meet these conditions through a high-quality pipeline comprising: (i) refined surface modeling with depth regularization for precise depth maps, and (ii) a perturbation-based view augmentation with pre-rendering checks to ensure fidelity. The following sections detail each component.

**Preliminaries of Gaussian Splatting:** 3DGS (Kerbl et al., 2023) explicitly reconstructs a 3D scene with millions of 3D Gaussian primitives $\{\mathcal{G}_i\}$, which are defined by a Gaussian function:

$$\mathcal{G}_i(\boldsymbol{x}|\boldsymbol{\mu}_i, \boldsymbol{\Sigma_i}) = e^{-\frac{1}{2}(\boldsymbol{x}-\boldsymbol{\mu}_i)^\top \boldsymbol{\Sigma}_i^{-1}(\boldsymbol{x}-\boldsymbol{\mu}_i)}, \tag{1}$$

where $\boldsymbol{\mu}_i \in \mathbb{R}^3$ and $\boldsymbol{\Sigma}_i \in \mathbb{R}^{3\times 3}$ are the center position and 3D covariance matrix, respectively. The covariance matrix $\boldsymbol{\Sigma}_i$ can be decomposed into a scaling matrix $\boldsymbol{S}_i \in \mathbb{R}^{3\times 3}$ and a rotation matrix $\boldsymbol{R}_i \in \mathbb{R}^{3\times 3}$ such that $\boldsymbol{\Sigma}_i = \boldsymbol{R}_i\boldsymbol{S}_i\boldsymbol{S}_i^\top \boldsymbol{R}_i^\top$. To render a pixel value $\boldsymbol{C} \in \mathbb{R}^3$ or a pixel depth $\boldsymbol{D} \in \mathbb{R}$, the primitives are first splatted to 2D, and rendering is performed as follows:

$$\boldsymbol{C} = \sum_i \boldsymbol{c}_i\alpha_i \prod_{j=1}^{i-1}(1-\alpha_j), \quad \boldsymbol{D} = \sum_i \boldsymbol{z}_i\alpha_i \prod_{j=1}^{i-1}(1-\alpha_j), \tag{2}$$

where $\alpha_i \in \mathbb{R}$ is calculated from a learned per-point opacity, $\boldsymbol{c}_i \in \mathbb{R}^3$ is the view-dependent color calculated from 3-degree Spherical Harmonics (SH), *i.e.* $\boldsymbol{sh} \in \mathbb{R}^{48}$, and $\boldsymbol{z}_i \in \mathbb{R}$ is the depth value in camera frame.

**Improving Depth Rendering for High-Precision Dense Labels.** $\alpha$**-blending** can be a common approach to obtain depth maps as shown in Eq. 2, namely computing an opacity-weighted average of the depths of all Gaussian primitives along each ray. $\alpha$-blending produces smooth depth maps but systematically biases geometry (shown in Fig. 1 (b-1)): the position of the surface is offset by opacity, and depth mixing artifacts occur near boundaries.

A simple but effective alternative is to identify the first **dominant primitive** along the ray whose opacity exceeds a threshold $\tau$ (to suppress floaters) and directly capture its depth value for the pixel:

$$\boldsymbol{D} = \boldsymbol{z}_k, \quad k = \min\{i \mid \alpha_i \geq \tau\}. \tag{3}$$

While this method avoids blending bias and yields more geometrically faithful depths, it introduces new defects: neighboring pixels snapping to the same primitive causes blocky surfaces (Fig. 1 (b-2)).

This motivates us to seek more refined surface reconstruction. A dominant primitive can be approximated by flattening each Gaussian ellipsoid into a plane along the camera's $z$-axis. Alternatively, compressing along the shortest axis yields a Gaussian plane that better preserves the ellipsoid shape. Specifically, following Chen et al. (2024a), we take the axis with the smallest scaling factor as the normal $n_i$ of the Gaussian plane, and apply $\alpha$-blending to render both the normal map $N$ and distance map $\mathcal{D}$:

$$N = \sum_{i \in N} R_c^T n_i \alpha_i \prod_{j=1}^{i-1}(1 - \alpha_j), \quad \mathcal{D} = \sum_{i \in N} d_i \alpha_i \prod_{j=1}^{i-1}(1 - \alpha_j), \tag{4}$$

where $R_c$ is the camera-to-world rotation, $\mu_i$ the Gaussian center, and $T_c$ the camera center. The plane-to-camera distance is $d_i = (R_c^T(\mu_i - T_c))^T(R_c^T n_i)$. The depth map is then obtained by ray-plane intersection:

$$D(p) = \frac{\mathcal{D}}{N(p)K^{-1}\tilde{p}}, \tag{5}$$

with pixel $p = [u, v]^T$, homogeneous coordinate $\tilde{p}$, and intrinsic $K$.

This fine-grained modeling yields smooth and accurate depth in well-covered regions (Fig. 1 (b-3)), but geometry degrades with sparse views. To address this, following Chung et al. (2024); Li et al. (2024), we scale monocular depth priors (Yang et al., 2024) with COLMAP (Schonberger & Frahm, 2016) and apply an $\ell_1$ loss to regularize rendered depth, enhancing rare-view quality and reducing floaters (Fig. 1 (b-4)).

**Novel-View Sampling and Pre-Rendering Check.** To generate novel views for image matching, we first define a set of camera projection matrices $\{P_i\}$, where $P_i = K_i[R_i|t_i]$ with intrinsic $K_i$, rotation $R_i$, and translation $t_i$. Using Eq. 2 and Eq. 5, we render both the color image $\{I_i\}$ and depth map $\{D_i\}$. Direct random sampling in a 3DGS scene often produces artifacts, incomplete regions, or unnatural perspectives, degrading data fidelity.

To alleviate this, we adopt a perturbation-based viewpoint generator that applies controlled jitters to training cameras. Specifically, $\Delta R$ and $\Delta t$ are sampled from a uniform distribution and added to extrinsics $[R|t]$, while a random scaling factor *scale* is applied to intrinsics $K$ to adjust $f_x, f_y$, simulating zoom-in/zoom-out variations.

To further guarantee quality, we perform *Pre-rendering Checks*. For each candidate viewpoint $v$, we first render its image $I_v$ and depth $D_v$ on-the-fly to compute statistical indicators $\Phi(v) = \{N_v, \bar{\alpha}_v, \rho_v^{valid}, \rho_v^{near}\}$, where $N_v$ is the number of contributing Gaussians, $\bar{\alpha}_v$ the average opacity, $\rho_v^{valid}$ the fraction of pixels exceeding opacity threshold $\tau_\alpha$, and $\rho_v^{near}$ the fraction below depth threshold $\tau_D$. For each metric $i \in \Phi$, we calculate its empirical mean $\mu_i$ and standard deviation $\sigma_i$ across candidates, and reject viewpoint $v$ if $|i(v) - \mu_i| > 2\sigma_i$. Only those passing all metrics are retained for final rendering and data generation.

## 3.2 REPRESENTATION ALIGNMENT

Our 3DGS-based data generation framework provides not only image pairs with dense correspondences but also the explicit 3D attributes (e.g., position, geometry, appearance) of Gaussian primitives. This allows us to reframe the core challenge of image matching: instead of matching ambiguous 2D pixel intensities, we are actually looking for projections of the same Gaussian primitive/cluster from different viewpoints.

To leverage this 3D information, we build upon ELoFTR (Wang et al., 2024) and LoFTR (Sun et al., 2021), strong transformer-based matchers whose semi-dense paradigm naturally aligns with the discrete nature of Gaussian projections. We observe that Gaussians correspond to image elements at multiple scales: individual primitives map to fine-grained pixels, while clusters (k-nearest neighbors or voxels) of primitives form local patches. This observation motivates our two complementary alignment strategies: (1) Coarse-level Representation Alignment: We align 2D patch features with aggregated 3D features derived from Gaussian clusters. (2) Fine-level Attribute Alignment: We enforce 3D geometric and appearance consistency directly on pixel-level matches.

**Preliminaries.** We concatenate the Gaussian center $\mu_i \in \mathbb{R}^3$, opacity $\alpha_i \in \mathbb{R}^1$, normalized scale factors $\tilde{s} \in \mathbb{R}^3$ (detailed in Appendix B.2), quaternion-based rotation $q_i \in \mathbb{R}^4$, and spherical har-

Figure 3: **Coarse-level representation alignment.** Given a coarse-to-fine matcher, local crops at 2D positions indicated by ground-truth coarse matches are encoded as patch embeddings. Simultaneously, 3D positions of the matches are used to query multi-scale voxel features from Point Transformer, which are encoded as voxel embeddings. Two embeddings are aligned via contrastive loss. The trained patch embedding head is then frozen and used to assist correlation computation.

monic coefficients $\boldsymbol{sh}_i \in \mathbb{R}^{48}$ into an explicit Gaussian feature $f_i^{\mathrm{gs}} \in \mathbb{R}^{59}$. For each image, we identify the Gaussian primitive that contributes the most to the pixel opacity in the rendering pipeline. At each pixel location, we record the index of its dominant Gaussian to construct a *Gaussian map* (denoted as $\mathbf{Map}^{GS}$), which enables subsequent queries of pixel-Gaussian relationships.

For common supervision of both fine-level and coarse-level matching stage, we follow Sarlin et al. (2020); Sun et al. (2021) to supervise the correlation score matrix $\mathcal{S}$ by minimizing the negative log-likelihood loss over ground-truth matches $\{\mathcal{M}\}_{gt}$, which are warped via pose and depth:

$$\mathcal{L} = -\frac{1}{N} \sum_{(\tilde{i},\tilde{j}) \in \{\mathcal{M}\}_{gt}} \log \mathcal{S}(\tilde{i}, \tilde{j}). \tag{6}$$

**Coarse-level representation alignment.** In the coarse matching stage, we aim to align the representations of 2D patches with the multi-scale 3D voxel/cluster representations aggregated from 3DGS, enabling the coarse matching to possess 3D awareness at the feature level. The following content is visualized in Fig. 3.

(1) **Patch Embedding:** To preserve the powerful 2D matching representations and minimize the influence of auxiliary alignment tasks on the main task, we augment the coarse feature maps with additional channels for 3D representation learning, denoted as $F_A^{3d}$ and $F_B^{3d}$. Meanwhile, the attention-transformed coarse feature maps are frozen and projected to a lower dimension to obtain $F_A^{2d}$ and $F_B^{2d}$. We then fuse $F^{2d}$ and $F^{3d}$ to obtain $F^{final}$. During training, we sample $N_{ps}$ ground-truth coarse matching points $\{\boldsymbol{p}_A^c, \boldsymbol{p}_B^c\}$, and crop a $3 \times 3$ region of the feature map around each point. Finally, a shared decoding head produces the patch embedding for each matching point. For the $i$-th matching pair, its patch embeddings in views $I_A$ and $I_B$ denoted as $\mathbf{q}_i^A, \mathbf{q}_i^B \in \mathbb{R}^{128}$. During inference, the $\mathbf{q}_i$ is concatenated into corresponding position in $F^{final}$, and the correlation matrix is computed to perform mutual nearest neighbor (MNN) matching.

(2) **Voxel Embedding:** During training, the ground-truth coarse points $\boldsymbol{p}_A^c$ are projected to 3D points $\boldsymbol{p}_{3d}^c$. The union of Gaussians in the two matching views, $f_A^{gs} \cup f_B^{gs}$, can be regarded as a featured point cloud. We employ PointTransformerV3 (Wu et al., 2024) to extract multi-scale voxel features $\{F_s^{\mathrm{voxel}} \mid s \in \{1, 1/2, 1/4, 1/8\}\}$ from this point cloud. For each $\boldsymbol{p}_{3d}^c$, we collect and concatenate its features across different voxel scales according to its coordinates, and then a shared decoding head produces a unique voxel embedding for each matching pair in 3D space, denoted as $\mathbf{v}_i \in \mathbb{R}^{128}$.

(3) **Embedding Alignment:** We employ InfoNCE loss to perform 2d-3d representation alignment. Specifically, all embeddings are first $L_2$-normalized. We treat the voxel embedding $\mathbf{v}_i$ as the anchor, and the corresponding patch embeddings $\{\mathbf{q}_i^A, \mathbf{q}_j^B\}$, as positive samples. All other voxel and patch embeddings corresponding to irrelevant targets within the same scene are treated as negative samples. Thus, anchor and positive samples are pulled closer while negative samples are pushed away in a self-supervised manner. We adopt the InfoNCE format as follows,

$$\ell_{voxel}(i) = -log \frac{exp(sim(\mathbf{v}_i, \mathbf{q}_i^A)/\tau) + exp(sim(\mathbf{v}_i, \mathbf{q}_i^B)/\tau)}{\sum_{\mathbf{z} \in \mathcal{Z}} exp(sim(\mathbf{v}_i, \mathbf{z})/\tau)}, \tag{7}$$

$$\ell_{patchA,B}(i) = -log \frac{exp(sim(\mathbf{q}_i^{A,B}, \mathbf{v}_i)/\tau) + exp(sim(\mathbf{q}_i^{A,B}, \mathbf{q}_i^{B,A})/\tau)}{\sum_{\mathbf{z} \in \mathcal{Z}} exp(sim(\mathbf{q}_i^{A,B}, \mathbf{z})/\tau)}, \tag{8}$$

$$\mathcal{L}_{voxel} = \frac{1}{N} \sum_{i=1}^{N} \ell_{voxel}(i), \quad \mathcal{L}_{patch} = \frac{1}{N} \sum_{i=1}^{N} \frac{\ell_{patchA}(i) + \ell_{patchB}(i)}{2}, \tag{9}$$

where $sim(\cdot)$ calculates dot product similarity, temperature $\tau > 0$ controls the sharpness of the similarity distribution. $\mathcal{Z}$ includes both positives and in-batch negatives. The final loss is obtained by averaging $\ell_{voxel}(i)$ and $\ell_{patch}(i)$ over all anchors and linearly combining them with weights $\lambda_v$ and $\lambda_q$, i.e., $\mathcal{L}_{InfoNCE} = \lambda_v \mathcal{L}_{voxel} + \lambda_q \mathcal{L}_{patch}$.

**Fine-level Direct Attribute Alignment.** Analogous to feed-forward 3DGS methods (Charatan et al., 2024; Chen et al., 2024b), we aim to enable the model to predict per-correspondence-aligned Gaussian attributes and constrain 3D geometric and appearance consistency directly on pixel-level matching. This is achieved through two complementary supervision signals. First, we introduce another negative log-likelihood loss as Eq. 6, namely Gaussian position loss, only for marked pixel pairs projected from the same Gaussian center. This encourages the network to anchor matches to the cores of the 3D primitives. Second, we enable the model to predict the underlying Gaussian attributes for each match through an attribute head. For a given fine-level match, we crop a local $3 \times 3$ patch from the feature map and decode the predicted Gaussian attributes $\{\hat{f}_i^{\mathrm{gs}}, \hat{f}_j^{\mathrm{gs}}\}$. We then supervise these predictions against the ground-truth attributes $\{f_i^{\mathrm{gs}}, f_j^{\mathrm{gs}}\}$, queried via the $\mathbf{Map}^{GS}$. This process applies an attribute loss consisting of $\ell_1$ regression and consistency terms, where the quaternion rotation $\boldsymbol{q}_i$ is represented by a 6D vector (Zhou et al., 2019) as intermediate form.

## 4 EXPERIMENTS

In this section, we conduct extensive experiments to validate our method. Sec. 4.1 evaluates the quality of ground-truth correspondences generated by our pipeline. Sec. 4.2 examines the zero-shot generalization of models trained on MatchGS across ScanNet (Dai et al., 2017a), MegaDepth (Li & Snavely, 2018), and ZEB (Shen et al., 2024). Sec. 4.3 presents ablations of key design choices, and Sec. 4.4 demonstrates performance on downstream tasks.

**Implementation Details.** We reconstruct 245 3DGS scenes from multi-view datasets including Mip-NeRF 360 (Barron et al., 2022), DeepBlending (Hedman et al., 2018), Tanks and Temples (Knapitsch et al., 2017), BungeeNeRF (Xiangli et al., 2022), DTU (Jensen et al., 2014), and DL3DV (Ling et al., 2024). Our pipeline then renders about 168K frames, maintaining a 1:1 ratio between train and augmented views (i.e., $1\times$ extra sampling), forming the MatchGS$_{245}$ training set. We also apply image augmentations to reduce the gap between rendered image and real image, including color jitter, random gamma adjustment, motion blur, and ISO noise. Details of data pipeline are provided in Appendix B. We use LoFTR (Sun et al., 2021) and its efficient variant ELoFTR (Wang et al., 2024) as baselines. Unless otherwise specified, both models are trained from scratch on the MatchGS$_{245}$ dataset, with our proposed representation alignment strategy applied as an additional self-supervision signal, resulting in the MATCHGS$_{\mathrm{ELoFTR}}$ and MATCHGS$_{\mathrm{LoFTR}}$ models. Further experimental details are provided in Appendix A.1.

### 4.1 DATA PIPELINE EVALUATION

We evaluate the accuracy of our generated correspondences using epipolar and relative reprojection error (see Appendix B.3 for metric details). We assume all methods obtain accurate poses. As shown in Tab. 1, 3DGS-based depth maps reduce epipolar error by 10 to $40\times$ compared to traditional methods, while their reprojection error lies between SfM- and depth-camera-based results. The Plane & Regularize variant performs best on both metrics, confirming that Plane Gaussians

| Method | Depth Source | Epi. ↓ | Std | Rel. ↓ | Std |
|---|---|---|---|---|---|
| $\alpha$-blending Depth | | $8.37 \times 10^{-6}$ | $7.44 \times 10^{-5}$ | 0.0293 | 0.0099 |
| Dominant Depth | 3DGS | $8.60 \times 10^{-6}$ | $8.19 \times 10^{-5}$ | 0.0203 | 0.0088 |
| Plane Depth | | $\mathbf{2.13 \times 10^{-6}}$ | $7.43 \times 10^{-6}$ | 0.0373 | 0.0109 |
| Plane & Regularize | | $2.35 \times 10^{-6}$ | $8.83 \times 10^{-6}$ | 0.0132 | 0.0082 |
| MegaDepth | SfM | $1.00 \times 10^{-4}$ | $2.45 \times 10^{-4}$ | 0.0498 | 0.0180 |
| ScanNet | Depth Camera | $1.01 \times 10^{-4}$ | $8.25 \times 10^{-4}$ | $\mathbf{0.0116}$ | 0.0083 |

Table 1: **Evaluations of epipolar error (Epi.) and relative reprojection error (Rel.).**

provide precise epipolar constraints. Moreover, adding depth regularization further improves depth consistency, approaching the quality of depth-camera methods.

Table 2: **Zero-shot or in-domain performance on ScanNet and MegaDepth** (↑). Methods trained with in-domain data are highlighted in orange ( : partial in-domain; : full in-domain). While our MATCHGS ensures totally zero-shot (in white).

| Method | AUC → | ScanNet-1500 | | | Mean | MegaDepth-1500 | | | Mean |
|---|---|---|---|---|---|---|---|---|---|
| | | @5° | @10° | @20° | | @5° | @10° | @20° | |
| SUPERGLUE (IN) | | **16.2** | **33.8** | **51.8** | **33.9** | 31.9 | 46.4 | 57.6 | 45.3 |
| SUPERGLUE (OUT) | | 15.5 | 32.9 | 49.9 | 32.8 | **42.2** | **61.2** | **76.0** | **59.8** |
| LoFTR (IN) | | 22.1 | 40.8 | 57.6 | 40.2 | 4.0 | 9.3 | 18.4 | 10.6 |
| LoFTR (OUT) | | 18.0 | 34.6 | 50.5 | 34.4 | 52.8 | 69.2 | 81.2 | 67.7 |
| ELoFTR (OUT) | | 19.2 | 37.0 | 53.6 | 36.6 | **56.4** | **72.2** | **83.5** | **70.7** |
| GIM$_{\text{LoFTR}}$ | | 19.5 | 37.3 | 55.1 | 37.3 | 51.3 | 68.5 | 81.1 | 67.0 |
| MATCHGS$_{\text{LoFTR}}$ | | 21.8 | 41.5 | 58.1 | 40.5 | 45.5 | 62.5 | 75.9 | 61.3 |
| MATCHGS$_{\text{ELoFTR}}$ | | **22.8** | **42.3** | **59.9** | **41.7** | 47.5 | 63.9 | 76.2 | 62.5 |
| DKM (IN) | | **29.4** | **50.7** | **68.3** | **49.5** | 59.2 | 74.1 | 84.7 | 72.7 |
| DKM (OUT) | | 26.4 | 46.6 | 63.7 | 45.6 | **60.4** | **74.9** | **85.1** | **73.5** |

Table 3: **Zero-shot performance on ZEB.** The four horizontal groups correspond to *handcrafted*, *sparse*, *semi-dense*, and *dense* methods. In the semi-dense group, the best results are bolded and the second-best underlined.

| Method | Mean Rank ↓ | Mean AUC@5°↑ | Real | | | | | | | | Simulate | | | |
|---|---|---|---|---|---|---|---|---|---|---|---|---|---|---|
| | | | GL3 | BLE | ETI | ETO | KIT | WEA | SEA | NIG | MUL | SCE | ICL | GTA |
| ROOTSIFT | 7.6 | 31.8 | 43.5 | 33.6 | 49.9 | 48.7 | 35.2 | 21.4 | 44.1 | 14.7 | 33.4 | 7.6 | 14.8 | 43.9 |
| SUPERGLUE (IN) | 10.3 | 21.6 | 19.2 | 16.0 | 38.2 | 37.7 | 22.0 | 20.8 | 40.8 | 13.7 | 21.4 | 0.8 | 9.6 | 18.8 |
| SUPERGLUE (OUT) | 7.3 | 31.2 | 29.7 | 24.2 | 52.3 | 59.3 | 28.0 | 28.2 | 48.0 | 20.9 | 33.4 | 4.5 | 16.6 | 29.3 |
| LoFTR (IN) | 10.6 | 10.7 | 5.6 | 5.1 | 11.8 | 7.5 | 17.2 | 6.4 | 9.7 | 3.5 | 22.4 | 1.3 | 14.9 | 23.4 |
| LoFTR (OUT) | 6.2 | 33.1 | 29.3 | 22.5 | 51.1 | 60.1 | 36.1 | 29.7 | 48.6 | 19.4 | 37.0 | 13.1 | 20.5 | 30.3 |
| ELoFTR (OUT) | 7.0 | 32.8 | 27.7 | 22.8 | 50.7 | 62.7 | 35.9 | 28.1 | 46.1 | 16.7 | 38.1 | 12.2 | 22.7 | 30.0 |
| GIM$_{\text{LoFTR}}$ | 4.7 | **39.1** | **50.6** | **43.9** | 62.6 | 61.6 | 35.9 | 26.8 | 47.5 | 17.6 | **41.4** | 10.2 | 25.6 | **45.0** |
| MATCHGS$_{\text{LoFTR}}$ | 5.1 | 36.8 | 35.8 | 29.6 | 61.4 | 63.9 | 35.2 | 27.9 | 48.6 | 21.5 | 38.7 | **13.2** | 24.2 | 41.8 |
| MATCHGS$_{\text{ELoFTR}}$ | **3.8** | 38.1 | 34.0 | 29.7 | **63.3** | **66.3** | **36.4** | **29.8** | **49.7** | **21.9** | 39.4 | 13.0 | **30.3** | 43.6 |
| DKM (IN) | 1.8 | 46.2 | 44.4 | 37.0 | 65.7 | 73.3 | 40.2 | 32.8 | 51.0 | 23.1 | 54.7 | 33.0 | 43.6 | 55.7 |
| DKM (OUT) | 1.5 | 45.8 | 45.7 | 37.0 | 66.8 | 75.8 | 41.7 | 33.5 | 51.4 | 22.9 | 56.3 | 27.3 | 37.8 | 52.9 |

## 4.2 ZERO-SHOT GENERALIZATION

**Results on MegaDepth and ScanNet benchmarks (Tab. 2).** Here some comparison methods use partial in-domain training (highlighted). On ScanNet, MATCHGS$_{\text{ELoFTR}}$ and MATCHGS$_{\text{LoFTR}}$ improve average AUC by 13.9% and 17.7% over outdoor baselines. Notably, MATCHGS$_{\text{LoFTR}}$, trained without in-domain data, outperforms GIM$_{\text{LoFTR}}$ (Shen et al., 2024), which use ScanNet as a training subset. Qualitative Results are shown in Fig. 4. On MegaDepth, although GIM$_{\text{LoFTR}}$, ELoFTR (out), and LoFTR (out) leverage in-domain data, our zero-shot method remains highly competitive. We further provide failed cases and analysis on MegaDepth in Appendix A.3, where severe illumination changes or extreme zoom-in causes matching failures, revealing potential future directions.

**Results on ZEB benchmark (Tab. 3).** Here all comparison methods follow the zero-shot protocol. MATCHGS$_{\text{ELoFTR}}$ and MATCHGS$_{\text{LoFTR}}$ achieve significant average AUC gains of 16.2% and 11.2%, respectively, showing strong competitiveness against GIM$_{\text{LoFTR}}$. Viewed from another angle, GIM is trained on a combination of reconstruction-based standard datasets and pseudo-labels from large-scale internet videos. While our method can serve as a new type of standard dataset with more precise geometry, richer viewpoints, and additional 3D information, thus complementing GIM and exploring a different direction for zero-shot training paradigms.

## 4.3 ABLATION STUDIES

We conduct ablation studies on the ScanNet test set using MATCHGS$_{\text{ELoFTR}}$ to evaluate the design choices in our data generation pipeline and representation alignment. As shown in Tab. 4, increasing either the sampling ratio or the number of scenes leads to clear improvements in AUC. However,

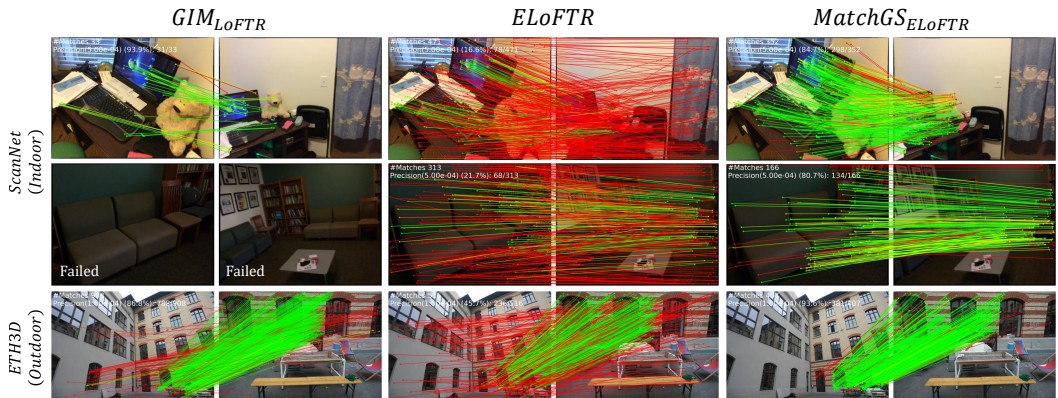

Figure 4: **Qualitative Results.** We compare with current state-of-the-art semi-dense matchers. Our method shows superior robustness under large viewpoint changes in both indoor and outdoor scenes.

Table 4: **Ablation Studies on data generation.** Table 5: **Ablation Studies on alignment strategy.**

| Condition | AUC | @5° | @10° | @20° |
|---|---|---|---|---|
| **Extra Sampling (70 Scenes)** | | | | |
| 0× Extra Sampling | | 19.0 | 37.2 | 54.2 |
| 1× Extra Sampling | | 21.2 | 40.1 | 57.5 |
| 2× Extra Sampling | | **22.4** | **41.7** | **59.2** |
| **Scenes Number (1× Extra)** | | | | |
| 70 Scenes | | 21.2 | 40.1 | 57.5 |
| 245 Scenes | | **22.8** | **42.3** | **59.9** |

| Method | AUC | @5° | @10° | @20° |
|---|---|---|---|---|
| Baseline | | 21.2 | 40.1 | 57.5 |
| **Coarse-level Representation Align.** | | | | |
| Intra-scene negatives | | **21.8** | **41.0** | **58.5** |
| Cross-scene negatives | | 21.5 | 40.5 | 57.8 |
| **Fine-level Attribute Align.** | | | | |
| Gaussian Position Loss | | 20.8 | 39.6 | 56.8 |
| Gaussian Position & Attribute Loss | | 20.5 | 39.4 | 56.8 |

while doubling the sampling ratio (2×) provides only marginal gains over 1×, it also doubles the storage cost. To balance performance and storage efficiency, we adopt 1× additional sampling as our final setting.

Tab. 5 compares the two representation alignment strategies. We find that coarse-level patch-to-voxel (or cluster) alignment consistently improves performance, yielding up to +0.6, +0.9, and +1.0 gains in AUC@5°, @10°, and @20°, respectively. This reveals that coarse-level representation can be stable and perceptually meaningful. Meanwhile, restricting negative samples in the InfoNCE loss to those within the same scene outperforms sampling across the entire batch (AUC@10° increases by 0.5), since it avoids penalizing embeddings of geometrically similar structures that appear in different scenes. In contrast, fine-level alignment with Gaussian position and attribute losses unexpectedly leads to performance degradation, with AUC@10° dropping by 0.7. This is likely because the attributes of individual Gaussian primitives are noisy and exhibit large variance across scenes. Such variance makes it difficult for the network to learn a stable pixel-to-primitive mapping.

### 4.4 DOWNSTREAM TASKS

We select MATCHGS$_{\text{ELoFTR}}$ for further evaluation on downstream tasks, including **homography estimation** on the HPatches dataset (Balntas et al., 2017) and **indoor/outdoor visual localization** on the InLoc (Taira et al., 2018) and Aachen v1.1 (Sattler et al., 2018) datasets. Without any fine-tuning, our model exhibits generalization in downstream tasks and shows better or similar performance than specialized models. Please refer to Appendix A.2 for our experiment results.

## 5 CONCLUSION

We propose MatchGS, a complete framework consisting of a data generation pipeline and a representation alignment strategy. It enhances the geometric quality of 3DGS to obtain diverse samples for zero-shot image matching and equips 2D matchers with viewpoint-invariant 3D perception. The significant zero-shot generalization shown in our experiments validates MatchGS as a promising and scalable alternative to traditional data paradigms, paving the way for more robust image matchers.

ETHICS STATEMENT

This work adheres to the ICLR Code of Ethics. Our research focuses on computer vision algorithms and does not involve human subjects, sensitive personal data, or potentially harmful applications. We believe that our dataset release and code contributions will benefit the community in a responsible and transparent manner.

REPRODUCIBILITY STATEMENT

We have made every effort to ensure the reproducibility of our work. Additional description of our dataset preparation process, as well as details of model training and hyperparameter configurations, is provided in Appendix A.1 and B. All data preprocessing and model training code has been submitted into an anonymous GitHub repository (available at: `https://github.com/anonymous186498/anonymous_code`). After the anonymity period, we will release our dataset, data generation toolbox, and training code, together with step-by-step tutorials to facilitate reproduction and further research.

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

## A  MORE EXPERIMENT DETAILS AND RESULTS

### A.1  MORE EXPERIMENT DETAILS

For ELoFTR (Wang et al., 2024) and LoFTR (Sun et al., 2021), their official outdoor models were trained on MegaDepth (Li & Snavely, 2018) (196 scenes) for 30 epochs, where 100 image pairs were randomly sampled from each sub-scene in every epoch (36,800 steps per epoch), amounting to about 1.1 million total training steps. To ensure a fair comparison, we align our training configuration on MatchGS$_{245}$ (245 scenes) with their outdoor model settings in terms of batch size, total training steps, and learning rate milestones. Specifically, we also sample 100 pairs from each sub-scene, resulting in 37,196 training steps per epoch. After training for 30 epochs, the model undergoes a total of 1,115,880 iterations. All other hyperparameters follow the original implementations, using gradient accumulation where necessary.

For the model input, we replace the original grayscale images with RGB images to align with the three-channel spherical harmonic coefficients of the Gaussian attributes. The model is trained with an input resolution of 832×832. Training on MatchGS is conducted using 4 NVIDIA RTX 3090 Ti GPUs, which takes approximately 3 days for ELoFTR and over 5 days for LoFTR.

### A.2  ADDITIONAL EXPERIMENT RESULTS

**Homography Estimation:** Following Dusmanu et al. (Dusmanu et al., 2019), we evaluate homography estimation on the HPatches dataset (Balntas et al., 2017) and report the area under the cumulative curve (AUC) of the corner error at thresholds of 3, 5, and 10 pixels. For fair comparison, we adopt the results reported in the original papers of competing methods. Compared to baseline approaches, MATCHGS$_{ELoFTR}$ achieves absolute improvements across all three metrics. Surprisingly, MATCHGS$_{ELoFTR}$ has also surpassed the dense matching method DKM (Edstedt et al., 2023).

Table 6: **Homography estimation.**

| Method AUC (%) → | @3px | @5px | @10px |
|---|---|---|---|
| SUPERGLUE (OUT) | 53.9 | 68.3 | 81.7 |
| LoFTR (OUT) | 65.9 | 75.6 | 84.6 |
| GIM$_{LoFTR}$ | 70.6 | 79.8 | 88.0 |
| ELoFTR (OUT) | 66.5 | 76.4 | 85.5 |
| MATCHGS$_{ELoFTR}$ | **71.4** | **80.7** | **88.8** |
| DKM (OUT) | 71.3 | 80.6 | 88.5 |

Table 7: **Indoor visual localization.** Unit: % of correctly localized queries (↑)

| Method | DUC1 | DUC2 |
|---|---|---|
| | (0.25m,10°) / (0.5m,10°) / (1.0m,10°) | |
| SUPERGLUE (IN) | 49.0 / 68.7 / 80.8 | 53.4 / 77.1 / 82.4 |
| LoFTR (IN) | 47.5 / 72.2 / 84.8 | 54.2 / 74.8 / 85.5 |
| ELoFTR (IN) | 52.0 / 74.7 / 86.9 | 58.0 / 80.9 / 89.3 |
| MATCHGS$_{ELoFTR}$ | 49.5 / 73.7 / 85.8 | 61.8 / 82.4 / 86.3 |
| DKM (IN) | 51.5 / 75.3 / 86.9 | 63.4 / 82.4 / 87.8 |

Table 8: **Outdoor visual localization.** Unit: % of correctly localized queries (↑).

| Method | Day | Night |
|---|---|---|
| | (0.25m,2°) / (0.5m,5°) / (1.0m,10°) | |
| SUPERGLUE (OUT) | 89.8 / 96.1 / 99.4 | 77.0 / 90.6 / 100.0 |
| LoFTR (OUT) | 88.7 / 95.6 / 99.0 | 78.5 / 90.6 / 99.0 |
| ELoFTR (OUT) | 89.6 / 96.2 / 99.0 | 77.0 / 91.1 / 99.5 |
| MATCHGS$_{ELoFTR}$ | 88.6 / 95.7 / 98.9 | 76.4 / 91.6 / 99.4 |
| DKM (OUT) | 84.8 / 92.7 / 97.1 | 70.2 / 90.1 / 97.4 |

**Visual Localization:** We further evaluate on two commonly used benchmarks, InLoc (Taira et al., 2018) and Aachen Day-Night v1.1 (Sattler et al., 2018), using the open-source HLoc framework (Sarlin et al., 2019) following prior work (Sun et al., 2021; Chen et al., 2022). For both datasets, we report the percentage of correctly localized queries under different pose error thresholds defined by angular and translational criteria, using results from the original papers of competing methods. On the indoor InLoc benchmark, MATCHGS$_{ELoFTR}$ attains similar or even better accuracy compared to ELoFTR (in) and LoFTR (in), which were trained on indoor data. On the outdoor Aachen v1.1 benchmark, MATCHGS$_{ELoFTR}$ achieves accuracy comparable to ELoFTR (out) and LoFTR (out), which were specifically trained for outdoor scenes. These results demonstrate the strong generalization ability and practical applicability of our method across diverse environments, without requiring scene-specific training.

### A.3 ANALYSIS OF FAILED CASES ON MEGADEPTH

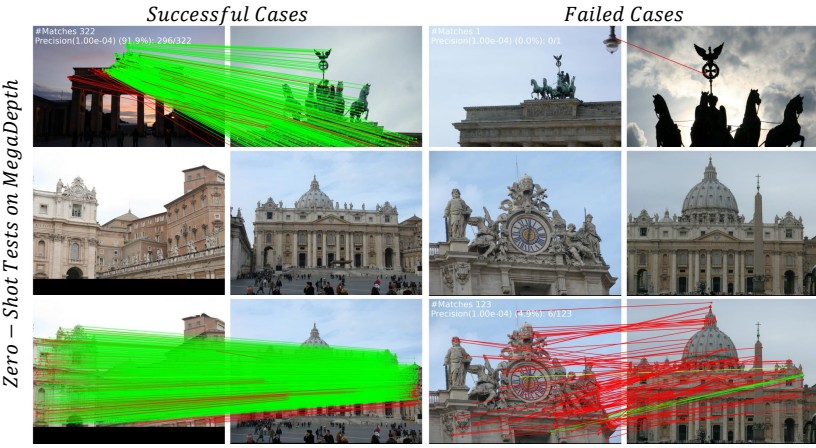

Figure 5: **Successful and failed cases on MegaDepth dataset.** Using MATCHGS$_{ELoFTR}$ for zero-shot testing.

As shown in Fig. 5, although our model can already handle some severe lighting changes and zoom-in scenarios in a zero-shot setting, it still fails under extreme lighting contrast (top right) and very large-scale zoom (bottom right). The failures under extreme lighting changes are likely due to the inability of our proposed data generation pipeline to simulate diverse real-world physical lighting,

which imposes limitations on the model in such conditions. Failures under very large-scale zoom arise from our restriction on the scaling factor of the focal length during data generation (excessive scaling can cause sampling artifacts). In this case, the zoom scale exceeds 6 times, while our maximum setting was 4 times, limiting the model's transfer performance.

# B DETAILS OF DATA PIPELINE AND DATASET

## B.1 DATA PROCESSING

Given a set of images from a multi-view dataset (all treated as training views), we first train a 3DGS scene using our geometry-improved framework. Next, for each training view, we generate several augmented viewpoints using a viewpoint generator. These augmented views are then processed with pre-rendering checks, removing a small number of low-quality views. Afterwards, the 3DGS renderer is used to produce the final images, depth maps, and Gaussian maps. Finally, we traverse all image pairs in the scene to compute their overlap and collect the image pair information used for training. For 245 scenes, the whole process takes about 2.5 days on 4 NVIDIA RTX 3090 Ti GPUs, with 80% of the time spent on 3DGS training.

## B.2 SCALE FACTOR NORMALIZATION FOR GAUSSIANS

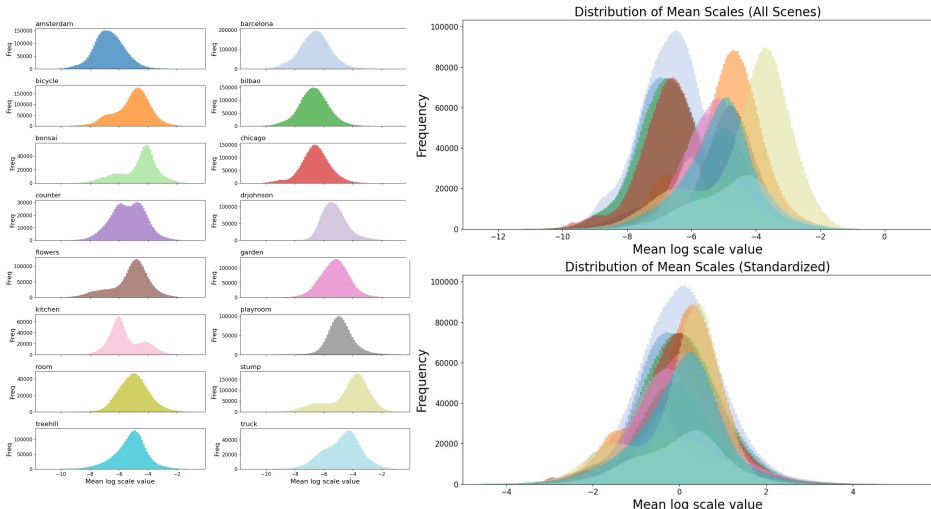

Figure 6: The distributions of the logarithm of mean scale factors across different 3DGS scenes. And the distributions after standardization.

In designing our representation alignment strategy we observe that scene scale varies dramatically between indoor and outdoor environments, and some reconstructed scenes do not possess a metric scale. This results in different magnitudes of **scale factors** for primitives in different scenes. While it introduces an ambiguity for learning a consistent 3D representation across scenes, which motivates a normalization of Gaussian scale factors across scenes. Let Gaussian primitives $\{\mathcal{G}_i\}$ in a scene have axis-aligned scale factors $s_{i,x}, s_{i,y}, s_{i,z}$. We define the per-primitive mean scale factor

$$s_i^{\mathrm{mean}} = \frac{s_{i,x} + s_{i,y} + s_{i,z}}{3}, \tag{10}$$

and work with the logarithm of scale factors. Denote $\ell_{i,k} = \log s_{i,k}, \ell_i^{\mathrm{mean}} = \log s_i^{\mathrm{mean}}$, where $k \in \{x, y, z\}$. According to our statistics shown in Fig., the distribution of $\ell_i^{\mathrm{mean}}$ within a scene can be well approximated by a Gaussian

$$\ell_i^{\mathrm{mean}} \sim \mathcal{N}(\mu, \sigma^2).$$

Thus, the Gaussian mean $\mu$ captures the overall scale factor magnitude of the scene. To remove the ambiguity introduced by different scene scales we standardize the per-axis log-scale factors by the

scene mean $\mu$. The standardized log-scale factors are computed as

$$\hat{\ell}_{i,k} \;=\; \frac{\ell_{i,k} - \mu}{\sigma}, \qquad k \in \{x, y, z\}. \tag{11}$$

In practice we estimate

$$\mu \;=\; \frac{1}{N} \sum_{i=1}^{N} \ell_i^{\mathrm{mean}} \quad \text{and} \quad \sigma \;=\; \sqrt{\frac{1}{N} \sum_{i=1}^{N} \left( \ell_i^{\mathrm{mean}} - \mu \right)^2},$$

where $N$ is the number of primitives in the scene. This normalization removes scene-level scale bias and reduces ambiguity when learning a shared 3D representation across scenes with widely differing and sometimes non-metric scales.

### B.3   EVALUATION METRICS FOR GROUND TRUTH

We first define the forms of epipolar error and relative reprojection error that we use in Sec 4.1. Let grid-sampled points (here we set grid size to 10 pixel) of two images be homogeneous $\tilde{\mathbf{x}} = [u, v, 1]^\top$ and $\tilde{\mathbf{x}}' = [u', v', 1]^\top$, and let $F$ be the fundamental matrix between the two views.

**Epipolar error.** The geometric epipolar error of a correspondence $(\mathbf{x}, \mathbf{x}')$ is the perpendicular distance from $\mathbf{x}'$ to the epipolar line $\mathbf{l}' = F\tilde{\mathbf{x}}$:

$$e_{\mathrm{epi}}(\mathbf{x}, \mathbf{x}') \;=\; \frac{\left|\tilde{\mathbf{x}}'^\top F \tilde{\mathbf{x}}\right|}{\sqrt{(F\tilde{\mathbf{x}})_1^2 + (F\tilde{\mathbf{x}})_2^2}}.$$

We use the symmetric version averages the distance in both directions:

$$e_{\mathrm{epi}}^{\mathrm{sym}} \;=\; \tfrac{1}{2} \left( \frac{\left|\tilde{\mathbf{x}}'^\top F \tilde{\mathbf{x}}\right|}{\|(F\tilde{\mathbf{x}})_{1:2}\|_2} + \frac{\left|\tilde{\mathbf{x}}^\top F^\top \tilde{\mathbf{x}}'\right|}{\|(F^\top \tilde{\mathbf{x}}')_{1:2}\|_2} \right).$$

**Relative Reprojection Error.** For points in the first image, we back-project each point to 3D, transform it to the second camera frame, and compute its projected depth $\hat{d}'$. Let $d'$ be the ground-truth depth at the corresponding pixel. The relative reprojection error is

$$e_{\mathrm{rel}} = \frac{1}{N} \sum_{i=1}^{N} \frac{|\hat{d}'_i - d'_i|}{d'_i}.$$

We next describe how we obtain the data in Tab. 1. For the four 3DGS-based methods, we randomly select 30 scenes for reconstruction and processed them through our data generation pipeline to obtain the dataset. For each scene, we randomly sample 100 image pairs (including training views and augmented views) such that the proportions of pairs with overlap ranges 0.1–0.3, 0.3–0.5, and 0.5–0.7 are 1:1:1. For MegaDepth (Li & Snavely, 2018) and ScanNet (Dai et al., 2017a), we follow the same procedure: 30 randomly selected scenes and 100 image pairs per scene, maintaining the same overlap distribution as above. Finally, for all sampled image pairs across datasets, we compute the epipolar error and relative reprojection error, reporting the mean and variance.

## C   LIMITATION AND FUTURE WORK

A limitation of our current work is the lack of lighting diversity in our data generation pipeline. As discussed in Appendix A.3, models trained with MatchGS are susceptible to failure under extreme illumination changes. However, we believe this can be addressed by incorporating recent 3DGS relighting techniques (Gao et al., 2024) into our pipeline, pointing to a valuable future direction in simulating harsh real-world conditions. Furthermore, our current training protocol samples image pairs of varying difficulty (e.g., different overlap levels) with uniform probability. Since our pipeline allows for active control over matching difficulty, another promising direction is to implement a curriculum learning (Bengio et al., 2009) strategy, progressing from easier to more challenging samples as training advances. Overall, we believe continued exploration of our 3DGS-based training framework holds significant potential for creating more robust and universal zero-shot image matchers.

## USE OF LARGE LANGUAGE MODELS

In preparing this paper, Large Language Models (LLMs) are used solely as auxiliary tools to assist with language polishing. The authors take full responsibility for all content written under their names, including any text that may have been refined with the aid of LLMs.

