# OpenReview forum: "Unlocking Zero-shot Potential of Semi-dense Image Matching via Gaussian Splatting"
_ICLR.cc/2026/Conference — ICLR 2026 Conference Withdrawn Submission_

### Official Review · Reviewer_JnBE · 2025-10-26

**Soundness:** 3
**Presentation:** 3
**Contribution:** 2
**Rating:** 2
**Confidence:** 5

**Summary:**

This work proposes MatchGS, which leverages 3D gaussian splatting (3DGS) to generate pseudo-labels for zero-shot image matching. The key idea is to first reconstruct the images using 3DGS, then remove outliers in the corresponding generation and render image pairs with matching pseudo-labels for training.

**Strengths:**

The idea of using 3DGS to generate image matching pseudo-labels is interesting.
The paper writing is clear.

**Weaknesses:**

1. The method largely follows the experiment setup of GIM, however, unlike GIM where the same method is applied to both sparse, semi-dense and dense matching methods, MatchGS only applies the proposed data to semi-dense methods.
- This raises the problem of generalization, since theoretically I dont see why the generated pseudo-labels cannot be applied to other method types, e.g., GIM only has sparse/semi-dense labels, it still can be applied to improve dense methods.
- This also raises the concern of the performance, since the current SOTA matching methods are all dense methods, the limited applicability also indicates that the proposed method might not be able to push the real SOTA.

2. Even in the semi-dense domain, the proposed method did not surpass the performance of GIM (see GIM_loftr and MatchGS+loftr) in table 3, which raises further concerns on the actual benefit given that GIM pipeline is much simpler (without the need of 3D reconstruction).

**Questions:**

NA

---

### Official Review · Reviewer_YAC1 · 2025-10-29

**Soundness:** 2
**Presentation:** 2
**Contribution:** 2
**Rating:** 2
**Confidence:** 3

**Summary:**

This paper proposes to use gaussian splatting networks to create unlimited amounts of paired correspondence data for training.

They first show that they can get good performance in comparison to other methods and then explore how the number of scenes impacts the results.

**Strengths:**

This paper proposes an interesting and scalable way to obtain paired data for correspondence training. In particular, they propose to use gaussian splatting in order to create high quality synthetic pairs and want to demonstrate that this means they have unlimited data with which to train a downstream model.

They compare a model trained with their approach to other methods for 2D correspondence, showing they get good results on 0-shot generalisation using MegaDepth and the ZEB dataset.

In order to leverage gaussian splatting, they train a model to give more robust correspondences and be robust to the noise that can be introduced by such an approach.

**Weaknesses:**

1. This paper does not quite fulfill its aims. The main aim is to see how using synthetic data can lead to unlimited data that can be used to demonstrate the value of such data. However, they generally seem to have fixed the amount of data to 70/245 scenes in Table 4. instead, it would be better to show data scaling -- how much do things improve as more and more scenes are sampled as opposed to a relatively small amount of 245 and only looking at two points. We would want to see that things continue to improve as the dataset gets bigger and bigger and we would a higher granularity to be able to see how quick gains are and how they plateau as we saturate at some data amount.

2. While they do a bit better than other semi-dense methods in the ZEB setup -- it is not consistent. And given that their point is the utility of their synthetic data -- I think they should have a graph showing how the amount of data impacts performance and then the performance of other methods to make it clear whether, with more data, we expect their setup to do consistently better or not. Finally, it would be interesting to understand if the improvements are the new points they find or the method -- could they train LoFTR / ELoFTR on their data and then compare results to what they have ?

3. The paper is not very clear for someone not familiar with the area
a.  -- why do they ignore dense methods -- I assume because they are slower but there is no computational comparison to be very convincing.
b. -- What is the AUC over in Table 2 / 3 -- I think pose but again not said ?

Given these two substantive weaknesses, I am not convinced that this paper is ready for acceptance.

**Questions:**

See above.

---

### Official Review · Reviewer_Kpbb · 2025-11-01

**Soundness:** 2
**Presentation:** 2
**Contribution:** 3
**Rating:** 2
**Confidence:** 4

**Summary:**

This paper addresses the problem of sparse matching datasets, which limits generalizability and restricts viewpoint distribution. They solve this by leveraging 3D Gaussian Splatting to render images from arbitrary user-specified camera trajectories without requiring hand-crafted annotations or sensor data. Through 3D-2D consistency matching, they enhance the matching network and demonstrate improved zero-shot matching performance compared to existing matchers.

**Strengths:**

- Utilizing 3D Gaussian Splatting to overcome the limitation of restricted camera trajectories in existing datasets is an interesting direction.
- To ensure geometric consistency in 3D Gaussian Splatting, introducing depth loss and planar Gaussians that enforce consistent geometry for matching is an intuitive and effective approach.
- By leveraging 3D Gaussians, they achieve additional 2D-3D consistency matching and propose injecting 3D information into the model.

**Weaknesses:**

### Weaknesses:

- **Unclear parameter ranges for camera perturbations:** What are the specific ranges of $\Delta R$ and $\Delta t$, and what scale is applied to the intrinsics? If these ranges are small, the claim that this dataset generation pipeline can produce extreme viewpoint changes may be overstated.
- **Concerns about 3DGS rendering quality:** I am curious about the rendering quality of the 3D Gaussian Splatting. To my knowledge, depth loss is sufficient in few-shot settings but may be inadequate for full-sequence settings because depth foundation models do not provide perfect depth estimates, and scale ambiguity can become problematic when processing full image sequences [1,2,3]. If the rendering quality is low, using synthetic datasets might be a more feasible approach for generating extreme viewpoint changes.
- **Potential for mixed dataset training:** I wonder whether there is a performance improvement when mixing the original dataset with your generated dataset. While I understand the table shows results for fair comparison, it would be valuable to see these results. Additionally, if performance improves with your dataset, would this approach also benefit dense matchers such as DKM[4] or RoMa[5], which show superior results compared to semi-dense matching methods?
- **Comparison with other augmentation methods:** How does this dataset generation approach compare to other augmentation paradigms[6] that create extreme viewpoint changes? Have alternative methods been explored that might yield better results?

### Minor Weakness:

- **Clarity of Equation (8):** The notation in Equation (8) is difficult to understand. Revising the q_{i}^{A,B} notation would improve readability.

### References
---
[1] Li, Jiahe, et al. "Dngaussian: Optimizing sparse-view 3d gaussian radiance fields with global-local depth normalization." Proceedings of the IEEE/CVF conference on computer vision and pattern recognition. 2024.

[2] Zhang, Chi, et al. "Hierarchical normalization for robust monocular depth estimation." Advances in Neural Information Processing Systems 35 (2022): 14128-14139.

[3] Chung, Jaeyoung, Jeongtaek Oh, and Kyoung Mu Lee. "Depth-regularized optimization for 3d gaussian splatting in few-shot images." *Proceedings of the IEEE/CVF Conference on Computer Vision and Pattern Recognition*. 2024.

[4] Edstedt, Johan, et al. "DKM: Dense kernelized feature matching for geometry estimation." *Proceedings of the IEEE/CVF Conference on Computer Vision and Pattern Recognition*. 2023.

[5] Edstedt, Johan, et al. "Roma: Robust dense feature matching." *Proceedings of the IEEE/CVF Conference on Computer Vision and Pattern Recognition*. 2024.

[6] Ma, Jiahao, et al. "Puzzles: Unbounded Video-Depth Augmentation for Scalable End-to-End 3D Reconstruction." *arXiv preprint arXiv:2506.23863* (2025).

**Questions:**

Questions are listed in the weakness section.

---

### Official Review · Reviewer_MX7F · 2025-11-01

**Soundness:** 2
**Presentation:** 1
**Contribution:** 2
**Rating:** 4
**Confidence:** 3

**Summary:**

The paper presents MatchGS, which aims to leverage 3DGS for generating data for image matching. MatchGS first adopts plane-based gaussians with additional depth regularization to learn geometrically consistent gaussians, and renders the scene in multiple views to generate image pairs with projected 3D gaussians serving as GT correspondence. Furthermore, the paper extends upon ELoFTR and LoFTR, by incorporating the information from 3D gaussian primitives. This is done in both coarse and fine levels, where the coarse level alignment aligns 2D features from the matching network and 3D features from a PointTransformerV3, while fine alignment adds a decoder head to predict the gaussian primitives from a local patch from the feature map.

**Strengths:**

1. The paper presents a self-supervised method, based on 3D gaussians, to generate data for training LoFTR-like models. The core motivation in this, being generating data in multiple views with gaussians for zero-shot performance, is very convincing.

2. The paper shows clear gains over LoFTR and ELoFTR despite being in a zero-shot setting, showing the effectiveness of proposed 3DGS-based learning of image matching.

**Weaknesses:**

1. The coarse and fine alignment strategy, despite being claimed complementary (L.265) does not show complementary results at all. In fact, the results for fine-level alignment drops in all cases and the method does not seem to incorporate it in its final design. If so, why is it introduced in the methodology section as a component?


2. Frankly, the paper is quite hard to follow. At first glance, it is unclear that how the GT correspondences are established for the generated data, where the reviewer assume the projected gaussian centers in two images to serve as the GT. The framework also has many components and loss functions, most of them only having brief justification and missing details.

**Questions:**

1. Although the authors find the gain to be "underwhelming" for extra sampling with the 2x results, it actually seems to be a quite significant margin. It is understandable that further scaling the number of samples can be limited by the storage that the data takes, but showing more scalability with the samples would greatly highlight the core motivation for the data generation pipeline. Could the authors show additional results with more samples, perhaps with some ways to reduce the storage usage?

---

### Note · Authors · 2025-11-12

I have read and agree with the venue's withdrawal policy on behalf of myself and my co-authors.